# Leveraging nonlinear relationships and interactions to improve 30-day pneumonia readmission machine learning models

Eric M. Mortensen*, Nkiruka Atuegwu, Shane J. Sacco

University of Connecticut School of Medicine, University of Connecticut Health Center, Farmington, Connecticut, United States of America

* mortensen@uchc.edu

## Abstract

### Background

Research is needed to develop more accurate readmission prediction models that identify patients at the highest risk of readmission after their initial pneumonia hospitalization. Improving prediction accuracy will support the implementation of more effective, personalized interventions to lower readmission rates. Published models tend to rely on traditional methods or advanced machine learning models that exclude continuous variables, overlooking opportunities to uncover nonlinear relationships and interactions. In response, we used electronic medical record (EMR) data, including continuous variables such as vitals, alongside more advanced machine learning (ML) models.

### Methods

Using EMR data from a single academic medical center, we identified adults with initial pneumonia admissions between April 2018 and February 2024. We predicted 30-day readmission using eXtreme Gradient Boosting (XGBoost) and deep neural networks, and compared their performance with that of traditional logistic regression.

### Results

We identified 2,752 patients admitted with pneumonia during the study period (mean age = 70.0 years, 49.1% female). The 30-day readmission rate was 9.9%. The average AUROC for our ML models ranged from 0.62 to 0.64, and AUPRC was 0.15 to 0.16, comparable to traditional logistic regression (0.63 and 0.17, respectively). Previously underemphasized predictors included drug abuse and BUN values.

**Data availability statement:** Our de-identified data is available at:https://www.kaggle.com/datasets/ssacco/nonlinear-relationships-and-pneumonia-readmissions.

**Funding:** This project was supported by the University of Connecticut CRISP program. The funders had no role in conducting the study or in the preparation, review, or approval of the manuscript.

**Competing interests:** The authors have declared that no competing interests exist.

## Conclusion

Using more advanced machine learning models and continuous variables yielded similar performance to logistic regression models. However, we identified previously understated predictors of readmission after pneumonia hospitalization. Future efforts should focus on gathering important data not readily available in EMR, such as social determinants of health, to potentially enhance the models.

## Introduction

Hospital readmissions among patients with pneumonia are frequent, costly, and potentially avoidable [1–5]. Despite efforts to optimize healthcare delivery, readmissions within 30 days are estimated to occur in up to 25% of patients hospitalized for pneumonia at a cost of $10 billion annually [2–5]. Since the Centers for Medicare and Medicaid Services (CMS) implemented the Hospital Readmission Reduction Program (HRRP) in 2012, there has been a focus on pneumonia readmissions because hospitals with higher-than-expected risk-adjusted 30-day readmission rates face major financial penalties [6].

Efforts to prevent readmissions include one or more interventions such as careful discharge planning, case management, self-management education, medication review, closely scheduled follow-up clinic appointments, phone calls, and home visits. However, results vary between institutions [7,8]. Specifically, some programs have been criticized insofar as being challenging to implement, resource-intensive, and largely unsustainable for most hospitals [9,10]. Despite nearly two decades of research, readmission rates remain alarmingly high throughout the United States. New approaches are essential. For example, targeting prospective interventions for inpatients labeled high risk of 30-day readmission by risk algorithms may prove effective and sustainable.

A recent systematic review showed that pneumonia readmission models performed poorly, with a median area under the ROC curve (AUROC) of 0.63 (range 0.59–0.77) [11]. They also used traditional modeling approaches, such as logistic regression (LR), which cannot automatically identify nonlinear relationships or interactions, unlike models like eXtreme Gradient Boosting (XGBoost) [12] or deep neural networks. Several studies have used more advanced machine learning (ML) models to predict readmission [13–16], but these models performed similarly to traditional methods, with a median AUROC of 0.65 (range 0.56–0.71). One reason for this similarity may be that these models excluded continuous variables, which could have benefited from nonlinear or interaction effects (e.g., vital signs or laboratory findings). Given that the data available in medical records (required for real-time prediction in healthcare settings) is generally limited, discovering these additional relationships could be even more important to model development. Thus, developing more advanced ML models with continuous variables may help drive the improvements necessary to identify additional patients at the highest risk of readmission and support the implementation of more effective, tailored, patient-centered interventions to reduce readmissions.

Our study aimed to use longitudinal electronic medical record data from a single academic medical center, along with more advanced ML models that incorporate not only typical binary predictors (e.g., historical diagnoses) but also continuous variables (e.g., vitals, laboratory results), to determine whether identifying patients at highest risk for readmission may be improved compared with the traditional modeling technique used in numerous published studies (LR), which cannot take advantage of automatic nonlinear and interaction modeling. Our a priori hypotheses were that we would (1) identify those at high risk for readmission better than LR models and (2) possibly identify novel factors associated with readmission after pneumonia that would otherwise be missed when linearly modeled.

## Methods

### Dataset and cohort

We extracted data from the Epic (Epic Systems Corporation) electronic health record platform for patients hospitalized at John Dempsey Hospital, an academic medical center in Farmington, Connecticut. Data included demographics, encounter details, medical history, diagnoses, medications, vital signs, and laboratory results. Geographic data were obtained from census tracts and linked to participants via zip code. Data was obtained on 3/5/2024, and the authors had access to identifiable data. Our cohort included all adult patients (>18 years) admitted to inpatient care from April 2018 to February 2024 who had a primary or secondary admission code for pneumonia due to infectious agents, identified using ICD-10 codes (J09-J20.0). The first pneumonia admission during the study period was designated as the index visit, and patients were followed for 30 days to determine whether they had a subsequent all-cause inpatient admission. We excluded patients who died during the index hospitalization or within 30 days of discharge without being readmitted (N = 229). The UConn Health Institutional Review Board classified this project as exempt research.

### Model features

We included features from various domains: demographics, past encounter information, comorbid conditions, medications, vital signs, laboratory results, and geographic demographics. *Demographic information* included age at the index visit, biological sex, insurance type (coded as Medicare, Medicaid, or other), language (coded as English, Spanish, or other), marital status (coded as single, married, divorced, or other), employment status (coded as not employed, employed, disabled, retired, or other), and Body Mass Index (calculated as weight in kg divided by height in meters squared). We collected self-reported race and ethnicity (coded as Black or African American, Hispanic or Latinx, White or Caucasian, or Other), but only used this information to describe patients due to concerns about model fairness and bias. *Encounter information* included the number of historical inpatient, ICU, outpatient, ED, and observation visits within one year of the index admission. Additionally, we included the length of the index visit in days.

*Comorbid conditions* included historical or present conditions at the index visit, identified by relevant ICD-10 codes and based on Charlson [17,18] and Elixhauser [19] indices (e.g., congestive heart failure). *Medications* encompassed outpatient prescriptions existing before hospital admission. *Vital signs* consisted of various measures recorded during admission and discharge of the index visit, including temperature, blood pressure, pulse, SpO2, and respiration.

*Laboratory results* included the first and last recorded values at the index visit for related tests. Specifically, we documented creatinine, BUN, hemoglobin, hematocrit, MCHC, sodium, white blood cell count, and whether hemoglobin A1C was measured. *Geographic demographics* included proportions of individuals with different ages, incomes, education levels, employment statuses, and homeownership. See S1 Table for a full list of variables.

### Model development

We predicted 30-day readmission using logistic regression (LR) and two more advanced machine learning (ML) models: eXtreme Gradient Boosting (XGBoost) [12] and deep neural network (DNN) models. We fine-tuned model

hyperparameters using 5-fold cross-validation to maximize the area under the ROC curve (AUROC). See S2 File for full details. To improve model generalizability and reduce overfitting, we reduced the feature space by performing an initial marginal screening procedure. Specifically, we tested each potential feature for prediction of 30-day readmission separately by conducting a series of Wilcoxon rank-sum tests or Fisher's exact tests, as appropriate. Features with $p < 0.10$, adjusted for false discovery rate using the Benjamini-Hochberg (BH) method, were then included in the modeling step.

## Model evaluation and description

To evaluate model performance, we randomly split our data into 80% training and 20% testing sets, stratifying by outcome status. After developing our models on the training set, we predicted patients' readmission status on the testing set and calculated AUROC and the area under the precision-recall curve (AUPRC). We also determined the sensitivity, positive predictive value, and specificity at the maximum Youden index (i.e., the optimal balance of sensitivity and specificity) [20]. This process was repeated 100 times, and we reported the average performance across all repetitions. To further evaluate discrimination and calibration, we visually examined calibration plots and calculated Brier scores (0 = perfect calibration and discrimination).

To describe our models, we reran them using the full cohort, summarizing key predictors and comparing characteristics of patients labeled high risk between models. First, to quantify predictor importance, we calculated SHAP (SHapley Additive exPlanations) values, a method derived from game theory that estimates the individual contribution of a predictor to overall risk scores across different levels of other predictors [21]. For binary predictors, we calculated SHAP values within patients who had that variable endorsed to highlight clinically important diagnoses. Most of these diagnoses occur only in a small subset of patients (e.g., < 10%). Their significance would be diminished or "washed out" by low prevalence in overall SHAP value calculations due to zero contributions from most patients. We presented average SHAP values across patients. Second, to better understand *which* patients were more likely to be identified as high risk by our models, we compared model feature prevalence or values between patients identified as high risk by each model using a series of Wilcoxon rank-sum tests (focused on unique risk labels) and McNemar tests; we again corrected *p*-values using the BH method. For this analysis, we set the high-risk threshold at the top decile of risk scores when creating the model with the full cohort. The alpha level for these two-sided tests was set at 0.05. All analyses were conducted in R version 4.5.1 [22].

## Results

Table 1 summarizes patient characteristics overall and by readmission status. Patients had a median age of 70.0 years and were approximately equally male (50.9%) and female (49.1%). Most patients were White or Caucasian (71.4%), covered by Medicare (62.8%), and married or in a relationship (40.3%). The median length of stay for the initial admission was 4.9 days. Regarding prior visits, more than half of the patients had outpatient visits (53.3%). Fewer patients had previous emergency visits (21.3%), inpatient admissions (11.8%), or visits requiring observation (3.7%). A total of 271 patients (9.9%) experienced a 30-day readmission for pneumonia following their initial admission. These patients did not differ from those who were not readmitted on most characteristics ($p > 0.09$), except for visit history. Specifically, readmitted patients were more likely to have had prior emergency visits (28.4% vs. 20.5% in non-readmitted; $p = 0.003$), prior inpatient admissions (24.7% vs. 10.4%; $p < 0.001$), and prior outpatient visits (63.1% vs. 52.3%; $p = 0.001$).

In Table 2, we summarize the performance of all models. For the LR model, the average AUROC was 0.63 (95% CI: 0.56–0.69), and the AUPRC was 0.17 (0.12–0.24). The average Youden index was 0.23 (0.14–0.34). At this threshold, the average sensitivity was 0.64 (0.32–0.92), with a positive predictive value of 0.16 (0.12–0.22) and a specificity of 0.59 (0.30–0.86). An average of 4% (0–10) of patients had a readmission in the lowest decile of risk scores, while 20% (13–28) did so in the highest. Our more advanced ML models performed similarly, providing comparable average metrics and mostly overlapping confidence intervals across all models. When visually inspecting calibration plots along with Brier scores, logistic regression seemed to have the best calibration according to the Brier score, although all models

**Table 1. Baseline patient characteristics.**

| Variable | Total cohort | | 30-day readmission | | | | p-value |
|---|---|---|---|---|---|---|---|
| | | | Yes | | No | | |
| **No of patients, % (n)** | 100.00 | (2,752) | 9.85 | (271) | 90.15 | (2,481) | |
| **Age, years, median (IQR)** | 70.00 | (56.00-83.00) | 72.00 | (57.00-83.00) | 70.00 | (56.00-83.00) | 0.25 |
| **Biological sex, % (n)** | | | | | | | 0.96 |
| Male | 50.94 | (1,402) | 51.29 | (139) | 50.91 | (1,263) | |
| Female | 49.06 | (1,350) | 48.71 | (132) | 49.09 | (1,218) | |
| **Race and ethnicity, % (n)** | | | | | | | 0.10 |
| Black or African American | 13.66 | (376) | 17.71 | (48) | 13.22 | (328) | |
| Hispanic or Latinx | 11.59 | (319) | 8.49 | (23) | 11.93 | (296) | |
| Other race or unknown | 3.38 | (93) | 2.95 | (8) | 3.43 | (85) | |
| White or Caucasian | 71.37 | (1,964) | 70.85 | (192) | 71.42 | (1,772) | |
| **Insurance type, % (n)** | | | | | | | 0.44 |
| Medicare | 62.75 | (1,727) | 65.68 | (178) | 62.43 | (1,549) | |
| Medicaid | 17.84 | (491) | 17.71 | (48) | 17.86 | (443) | |
| Other | 19.40 | (534) | 16.61 | (45) | 19.71 | (489) | |
| **Marital status, % (n)** | | | | | | | 0.23 |
| Married/in relationship | 40.30 | (1,109) | 38.38 | (104) | 40.51 | (1,005) | |
| Divorced/separated | 12.57 | (346) | 14.39 | (39) | 12.37 | (307) | |
| Single | 25.91 | (713) | 29.52 | (80) | 25.51 | (633) | |
| Other | 21.22 | (584) | 17.71 | (48) | 21.60 | (536) | |
| **Index encounter stay length, days, median (IQR)** | 4.87 | (3.06-7.30) | 5.09 | (3.14-8.06) | 4.85 | (3.05-7.21) | 0.09 |
| **Clinical utilization history, % (n)** | | | | | | | |
| Emergency | 21.26 | (585) | 28.41 | (77) | 20.48 | (508) | 0.003 |
| Inpatient | 11.77 | (324) | 24.72 | (67) | 10.36 | (257) | <0.001 |
| Observation | 3.67 | (101) | 4.06 | (11) | 3.63 | (90) | 0.85 |
| Outpatient | 53.34 | (1,468) | 63.10 | (171) | 52.28 | (1,297) | 0.001 |

performed relatively similarly. Notably, from calibration plots (data not shown), all models consistently predicted higher probabilities of readmission than actually observed across the risk spectrum.

In Fig 1, we show the top ten most influential model predictors, ranked by absolute SHAP values. The most significant predictors in LR models included first and last hemoglobin and hematocrit levels (top 4 predictors), as well as a drug abuse diagnosis, although only a small number of patients had this diagnosis (1.1%; n = 30). The XGBoost model also highlighted the importance of a drug abuse diagnosis (top predictor), along with fluid and electrolyte, rheumatoid, and pulmonary disorder diagnoses. The DNN model identified first and last BUN values and last pulse as top predictors, followed by last hematocrit and fluid and electrolyte disorders. Given that we calculated binary variable SHAP values only within patients who endorsed that variable, we then recalculated global SHAP values without this restriction (S3 File). As expected, almost all diagnoses dropped out of the list of most important predictors and were replaced by continuous variables that were present in all patients. This was an exception for employment status in the logistic regression model and for fluid/electrolyte disorders in the XGBoost model. The top predictors within the DNN model remained largely unchanged since they were already primarily continuous variables.

Table 3 summarizes differences in model feature prevalence or values between patients labeled high risk by the LR versus ML models. Overall, those labeled high risk by the ML models were more likely to have almost every comorbidity than those labeled by the LR model (e.g., renal disease: LR 7.6%, XGBoost 26.1%, DNN 40.6%). Additionally, ML-labeled patients were

**Table 2. Average performance of models in testing splits.**

| Performance, M (95% CI) | Model | | |
|---|---|---|---|
| | Logistic regression | XGBoost | Deep neural network |
| **AUROC** | 0.63 (0.56-0.69) | 0.62 (0.55-0.68) | 0.64 (0.58-0.70) |
| **AUPRC** | 0.17 (0.12-0.24) | 0.16 (0.11-0.21) | 0.15 (0.08-0.23) |
| **Brier score** | 0.09 (0.08-0.09) | 0.12 (0.09, 0.25) | 0.12 (0.10-0.15) |
| **Proportion** | 0.43 (0.16-0.72) | 0.39 (0.12-0.69) | 0.41 (0.15-0.75) |
| **Youden Index** | 0.23 (0.14-0.34) | 0.23 (0.14-0.33) | 0.25 (0.16-0.33) |
| **Sensitivity** | 0.64 (0.32-0.92) | 0.60 (0.27-0.89) | 0.64 (0.31-0.95) |
| **PPV** | 0.16 (0.12-0.22) | 0.16 (0.12-0.23) | 0.16 (0.12-0.23) |
| **Specificity** | 0.59 (0.30-0.86) | 0.63 (0.33-0.89) | 0.61 (0.27-0.87) |
| **High-risk patients** | | | |
| Bottom decile | 0.04 (0.00-0.10) | 0.05 (0.01-0.11) | 0.03 (0.00-0.09) |
| Top decile | 0.20 (0.13-0.28) | 0.19 (0.11-0.26) | 0.19 (0.13-0.27) |

Notes: AUROC = area under the ROC curve. AUPRC = area under the precision-recall curve. PPV = positive predictive value. XGBoost = extreme gradient boosting model.

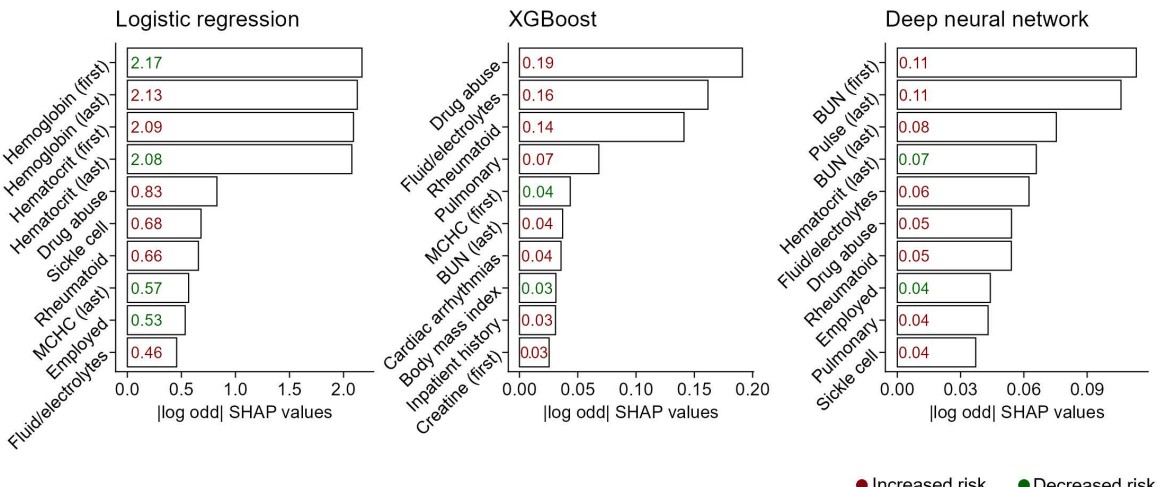

**Fig 1. SHAP (SHapley Additive exPlanations) values of model predictors.** This figure contains three bar plots displaying SHAP values in log odds (x-axis) for the top ten predictors of each model (y-axis). The left, middle, and right plots represent results for the logistic regression model, XGBoost model, and deep neural network, respectively.

more likely to have all types of historical encounters than LR-labeled patients (e.g., all outpatient: median LR 1.0, XGBoost 4.0, DNN 3.0). All laboratory findings also differed significantly. Hematocrit and hemoglobin levels were higher among LR-labeled patients compared to ML-labeled patients. Notably, BUN values varied greatly across models, being lowest in LR-labeled patients (first median: 18.0) and abnormally high in DNN-labeled patients (median 45.0). All *p*-values were less than 0.001.

## Post hoc analysis

We hypothesized that more advanced machine learning models may uncover nonlinear relationships when continuous variables are included as predictors. First and last BUN values emerged as top predictors in the deep neural network.

**Table 3. Differences in model features in labeled high-risk patients by model.**

| | Logistic regression | XGBoost | Deep neural network |
|---|---|---|---|
| Model candidates | N (%) or median (IQR) | N (%) or median (IQR) | N (%) or median (IQR) |
| **Demographic information** | | | |
| Body mass index | 27.37 (23.92-31.16) | 27.12 (21.80-27.91)** | 27.37 (22.62-27.40)** |
| Employed | 34 (12.32) | 23 (8.33) | 13 (4.71)** |
| **Vital signs** | | | |
| Temperature (first) | 100.00 (97.50-100.50) | 97.90 (97.00-100.23)*** | 97.90 (97.00-100.20)*** |
| Pulse (last) | 71.00 (62.00-82.00) | 75.00 (64.00-100.00)*** | 73.50 (64.00-84.00)*** |
| **Laboratory findings** | | | |
| Creatine (first) | 0.90 (0.80-1.30) | 1.10 (0.80-2.42)*** | 2.00 (1.30-3.12)*** |
| Creatine (last) | 0.80 (0.70-1.10) | 0.95 (0.70-1.72)*** | 1.40 (0.90-2.55)*** |
| BUN (first) | 18.00 (13.00-26.00) | 25.00 (14.00-42.25)*** | 45.00 (30.00-64.25)*** |
| BUN (last) | 18.00 (12.00-25.00) | 21.00 (13.00-44.00)*** | 36.50 (22.00-52.00)*** |
| Hemoglobin (first) | 12.00 (10.47-13.40) | 9.30 (7.88-10.90)*** | 9.30 (8.20-10.60)*** |
| Hemoglobin (last) | 11.30 (10.00-12.83) | 9.05 (8.00-10.40)*** | 9.00 (8.00-10.03)*** |
| Hematocrit (first) | 37.05 (32.80-40.70) | 29.95 (24.85-35.35)*** | 29.40 (26.25-33.15)*** |
| Hematocrit (last) | 35.25 (32.08-39.70) | 29.40 (25.70-33.73)*** | 28.60 (25.60-32.12)*** |
| MCHC (first) | 32.40 (31.37-33.50) | 31.60 (30.00-33.20)*** | 31.85 (30.67-32.80)*** |
| MCHC (last) | 32.30 (31.40-33.23) | 31.60 (30.10-32.70)*** | 31.70 (30.80-32.60)*** |
| **Diagnoses** | | | |
| Rheumatoid | 5 (1.81) | 17 (6.16)* | 10 (3.62) |
| Renal disease | 21 (7.61) | 72 (26.09)*** | 112 (40.58)*** |
| Congestive heart failure | 30 (10.87) | 85 (30.80)*** | 96 (34.78)*** |
| Cardiac arrythmia | 37 (13.41) | 84 (30.43)*** | 90 (32.61)*** |
| Pulmonary/circulatory | 4 (1.45) | 31 (11.23)*** | 25 (9.06)*** |
| Hypertension | 15 (5.43) | 85 (30.80)*** | 103 (37.32)*** |
| Chronic pulmonary | 36 (13.04) | 91 (32.97)*** | 73 (26.45)*** |
| Renal failure | 21 (7.61) | 72 (26.09)*** | 112 (40.58)*** |
| Fluid/electrolytes | 23 (8.33) | 110 (39.86)*** | 95 (34.42)*** |
| Drug abuse | 5 (1.81) | 15 (5.43)* | 14 (5.07)* |
| Acute kidney disease | 9 (3.26) | 44 (15.94)*** | 61 (22.10)*** |
| Nicotine dependence | 6 (2.17) | 22 (7.97)** | 19 (6.88)** |
| Sickle cell | 13 (4.71) | 48 (17.39)*** | 34 (12.32)** |
| **Medications** | | | |
| Anti-coagulants | 19 (6.88) | 46 (16.67)*** | 54 (19.57)*** |
| **Historical encounters** | | | |
| Emergency | 0.00 (0.00-0.00) | 0.00 (0.00-1.00)*** | 0.00 (0.00-1.00)*** |
| Inpatient | 0.00 (0.00-0.00) | 1.00 (0.00-1.00)*** | 0.00 (0.00-1.00)*** |
| Specialty | 0.00 (0.00-2.00) | 2.00 (0.00-6.00)*** | 1.00 (0.00-5.00)*** |
| All outpatient | 1.00 (0.00-5.00) | 4.00 (1.00-13.00)*** | 3.00 (0.00-11.00)*** |

Notes: *p < 0.05. **p < 0.01. ***p < 0.001.

While we cannot learn the *true* function linking BUN values to readmission risk (i.e., the "black box" problem), we wanted to explore whether the relationship between BUN values and deep neural network risk scores might be nonlinear. In Fig 2, we visualize predictions from two spline regression models that predict risk scores using (1) first BUN and (2) last BUN.

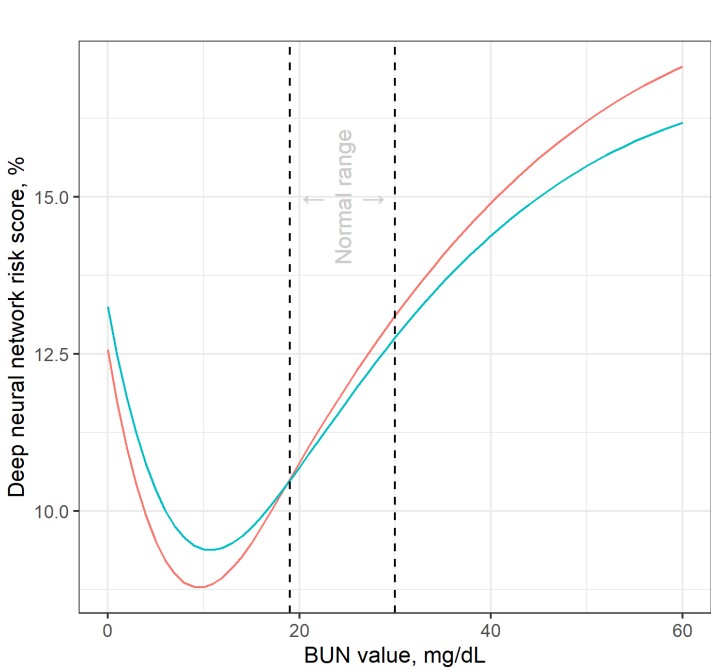

**Fig 2. First and last BUN values by deep neural network risk scores.** This figure is a line graph of the relationships between deep neural network risk scores (y-axis) and first and last BUN values (x-axis; red and blue, respectively). Two vertical reference lines (dashed) represent the normal range of BUN values (19-30 mg/dL).

We used clinical cut-offs of <19 and >30 mg/dL for BUN values as spline knots. It did indeed appear that the relationship between BUN values and risk was nonlinear. Lower BUN values were associated with higher risk scores when BUN was < 10 mg/dL. For ≥10 mg/dL, risk increased with BUN values, and the linearity of that increase appeared to slightly taper off at much higher values (e.g., from 40–60 mg/dL).

In this post hoc analysis, we found that the relationship between BUN values and risk scores from the deep neural network was nonlinear. We cannot confirm the *true* function of this relationship, as the nonlinearity may have been due to the interplay among multiple predictors. However, these results provide some empirical evidence that the model may have uncovered a nonlinear relationship, highlighting the importance of abnormally low *or* high BUN values in predicting readmission.

To help improve the generalizability of our models, we carried out an initial marginal screening process, which involves a tradeoff because it may miss variables that influence importance only through interactions. Since XGBoost and DNN models can evaluate feature importance natively and still consider variables important only in interactions, we reran our primary analyses without the marginal screening process (S4 File).

In this post hoc analysis, we observed that the performance of all models generally stayed the same, although their metrics were slightly worse. The logistic regression model was affected the most, with the average AUROC decreasing from 0.63 to 0.59, and the percentage of patients with readmissions in the bottom decile of risk scores increasing from 4% to 9%.

## Discussion

Our study compared LR and ML models for predicting 30-day readmissions after pneumonia hospitalization, using extensive longitudinal electronic medical record data from an academic medical center. All models achieved AUROCs between

0.62 and 0.64, matching the range of previously published pneumonia readmission models using traditional methods (median 0.63) [11] and more advanced ML methods (median 0.65) [13–16]. Because these studies (and ours) have examined different age groups, included comprehensive longitudinal EMR data and features, and used more advanced models, predictive performance of pneumonia readmission models may be approaching a ceiling, regardless of cohort design, EMR scope, or model complexity. Importantly, however, our study identified several predictors that had not been previously emphasized in pneumonia readmission research.

All of our models identified substance abuse as an important predictor of readmission among the few patients with this diagnosis code (1.1%); for the overall sample, vitals and laboratory findings were most important based on global SHAP values. Although this diagnosis has appeared in past work [23], its significance to the models was minimal. This highlights the vulnerability of patients with substance use disorders, who may face many healthcare challenges that increase their chances of readmission. Although substance abuse has been recognized as a general risk factor for hospital readmissions across various conditions [24], its role as a top predictor in pneumonia-specific models is a new insight that deserves further research.

Our advanced ML models, especially DNNs, identified first and last BUN levels as top predictors, which may affect clinical care for patients hospitalized with pneumonia. Our analysis showed that patients labeled as high risk by LR models had slightly lower median BUN values (18.0), whereas those identified by the DNN had notably higher values (46.0). Subsequently, it appears that BUN levels outside the normal range may not only indicate underlying conditions that increase pneumonia severity during index visits but also the risk of readmission. Low BUN levels may suggest malnutrition, liver dysfunction, or fluid overload, whereas high levels could point to kidney dysfunction, dehydration, upper gastrointestinal bleeding, or increased protein breakdown. The prevalence of BUN abnormalities in our models underscores the need to carefully monitor renal function, nutritional status, and fluid balance during pneumonia hospitalizations.

Despite modest predictive performance, our findings have several clinical implications. First, we developed and evaluated multiple machine learning methods (XGBoost, DNN) that leverage the nonlinearity of continuous variables. However, the consistent performance of our models and other published studies suggests that the modest predictive accuracy may be due to the inherent limitations of the data available in electronic medical records. EMRs lack key social determinants of health (e.g., housing stability, health literacy, lifestyle habits, environmental factors), which can affect readmission risk. Therefore, collecting some of this information during the intake process could improve the algorithms.

Additionally, consistent results across different methods and studies may reveal that important underlying phenomena could occur after discharge (e.g., complications, poor medication adherence). In this context, following up with patients by phone and offering a brief survey or interview on these topics can help refine risk assessments after initial algorithm predictions and potentially prevent readmissions.

Third, the low positive predictive values of the models (0.15–0.17) at the optimal balance of sensitivity and specificity indicate that, even at our best performance, the model would classify many patients as high-risk who will not actually be readmitted. This has important implications for resource allocation and intervention targeting, as overly broad efforts might not be cost-effective or sustainable. Healthcare providers can implement less resource-intensive responses, such as providing written materials (e.g., pamphlets), having a brief conversation at discharge (e.g., emphasizing the importance of post-discharge lifestyle habits and medication adherence), or conducting a follow-up phone call within 30 days to monitor patient health or concerns before a readmission occurs. Importantly, the risk gradient from the lowest to highest decile (3–5% to 19–20% readmission rate) demonstrates meaningful risk stratification that could inform clinical decision-making; the only question left is the degree of clinical response to that stratification.

Although we are hesitant to provide specific guidelines for implementation since these algorithms are not currently used by any hospitals and logistics, such as resource allocation and workflow integration, remain uncertain, these models could still be used for both risk stratification and resource allocation. For example, labeling nearly half of the sample as high risk (39–43% at the Youden Index) resulted in PPVs between 0.15 and 0.17. These patients could have brief discharge

conversations. Labeling 10% of the sample resulted in PPVs between 0.19 and 0.20. These patients could also receive follow-up phone calls.

Several important limitations must be acknowledged. First, this is a single-center study conducted at an academic medical center, which may limit the generalizability of the results to other healthcare settings (e.g., small rural hospitals). Second, the lower readmission rate observed in our cohort compared with national averages might reflect unique characteristics of our institution or patient population, potentially affecting the models applicability elsewhere. Third, our outcome measure—all-cause readmission—includes admissions unrelated to the index pneumonia hospitalization, which may introduce noise into our models. For example, some patients may be readmitted for unrelated injuries (e.g., falls, accidents) rather than for sequelae of pneumonia. In such cases, admission characteristics reflecting pneumonia severity would not be informative. Consequently, the modest performance metrics (e.g., AUROC, AUPRC) may be partly attributable to a misalignment between admission characteristics and unrelated readmissions in some patients. However, this approach aligns with CMS quality measures and captures real-world clinical concerns about any readmission following pneumonia discharge. Previous studies have shown that a minority of readmissions are due to pneumonia-related factors [25]. Finally, we lightly screened our features to help trained models better generalize to the testing data. When removing this step in our post hoc analysis, it seemed particularly important for the logistic regression model, while the advanced ML models showed little change. We recognize that more sophisticated screening methods exist that could have affected our results, which remains a limitation of this work.

Future research should focus on several important areas. First, adding model features such as social drivers of health (e.g., social support) and other patient-reported outcomes may enhance predictive accuracy. Second, combining EMR data with community resources, pharmacy databases, and social service records could also provide a more complete view of readmission risk. Third, exploring methods that integrate multiple data sources and prediction techniques might yield better results than standalone algorithms (e.g., data linkage, transfer learning, ensemble modeling). Fourth, hybrid strategies that incorporate clinician judgment, social worker assessments, and algorithmic predictions may help address some limitations of purely data-driven models.

In conclusion, our study shows that combining comprehensive EMR data with machine learning methods can identify new predictors of readmission after hospitalization for pneumonia, such as substance abuse and BUN levels. However, prediction accuracy remains modest across various models, indicating that challenges in predicting readmission extend beyond algorithmic limitations. These results underscore the complexity of post-discharge outcomes and highlight the need for multifaceted approaches to reduce readmissions, including thorough care coordination, social support, and targeted strategies for high-risk groups. Developing practical decision support tools that integrate prediction with intervention guidance may ultimately be more effective than solely aiming to improve prediction accuracy.

## Supporting information

**S1 Table. Study variables.** This table contains all study variables considered for use as model features. The left column is the category (e.g., demographics). The right column is the individual variable (e.g., patient sex).
(DOCX)

**S2 File. Model hyperparameter tuning details.** This file contains detailed information regarding the development of XGBoost and deep neural network models. Specifically, the training procedure and hyperparameter ranges utilized for development are provided.
(DOCX)

**S3 File. Global SHAP values.** This file presents results from calculating SHAP values without requiring calculations for binary variables to be within patients that endorsed that binary variable.
(DOCX)

**S4 File. Running models without the marginal screening process.** This file provides summarized results of a post hoc analysis where primary analyses were conducted without the initial marginal screening process. This file also includes a table of performance metrics.
(DOCX)

## Author contributions

**Conceptualization:** Nkiruka Atuegwu.

**Data curation:** Eric M. Mortensen, Nkiruka Atuegwu.

**Formal analysis:** Nkiruka Atuegwu, Shane J. Sacco.

**Funding acquisition:** Eric M. Mortensen, Nkiruka Atuegwu.

**Investigation:** Eric M. Mortensen, Shane J. Sacco.

**Methodology:** Eric M. Mortensen, Shane J. Sacco.

**Project administration:** Eric M. Mortensen.

**Supervision:** Eric M. Mortensen.

**Validation:** Shane J. Sacco.

**Visualization:** Shane J. Sacco.

**Writing – original draft:** Eric M. Mortensen.

**Writing – review & editing:** Nkiruka Atuegwu, Shane J. Sacco.

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
