## [Decision Letter · Decision Letter 0]

17 Mar 2026

PONE-D-26-02589Leveraging Nonlinear Relationships and Interactions to Improve 30-Day Pneumonia Readmission Machine Learning ModelsPLOS One

Dear Dr. Mortensen,

Thank you for submitting your manuscript to PLOS ONE. After careful consideration, we feel that it has merit but does not fully meet PLOS ONE’s publication criteria as it currently stands. Therefore, we invite you to submit a revised version of the manuscript that addresses the points raised during the review process.

**Please revise according to the reviewer's comments.**

We look forward to receiving your revised manuscript.

Kind regards,

Oliver Schildgen

Academic Editor

PLOS One

**Journal Requirements:**

1. When submitting your revision, we need you to address these additional requirements. Please ensure that your manuscript meets PLOS ONE's style requirements, including those for file naming. The PLOS ONE style templates can be found at https://journals.plos.org/plosone/s/file?id=wjVg/PLOSOne_formatting_sample_main_body.pdf and https://journals.plos.org/plosone/s/file?id=ba62/PLOSOne_formatting_sample_title_authors_affiliations.pdf 2. Please note that PLOS One has specific guidelines on code sharing for submissions in which author-generated code underpins the findings in the manuscript. In these cases, all author-generated code must be made available without restrictions upon publication of the work. Please review our guidelines at https://journals.plos.org/plosone/s/materials-and-software-sharing#loc-sharing-code and ensure that your code is shared in a way that follows best practice and facilitates reproducibility and reuse. 3. We note that the grant information you provided in the ‘Funding Information’ and ‘Financial Disclosure’ sections do not match.  When you resubmit, please ensure that you provide the correct grant numbers for the awards you received for your study in the ‘Funding Information’ section. 4. Thank you for stating the following financial disclosure: UConn Health CRISP grant   Please state what role the funders took in the study.  If the funders had no role, please state: "The funders had no role in study design, data collection and analysis, decision to publish, or preparation of the manuscript." If this statement is not correct you must amend it as needed. Please include this amended Role of Funder statement in your cover letter; we will change the online submission form on your behalf. 5. When completing the data availability statement of the submission form, you indicated that you will make your data available on acceptance. We strongly recommend all authors decide on a data sharing plan before acceptance, as the process can be lengthy and hold up publication timelines. Please note that, though access restrictions are acceptable now, your entire data will need to be made freely accessible if your manuscript is accepted for publication. This policy applies to all data except where public deposition would breach compliance with the protocol approved by your research ethics board. If you are unable to adhere to our open data policy, please kindly revise your statement to explain your reasoning and we will seek the editor's input on an exemption. Please be assured that, once you have provided your new statement, the assessment of your exemption will not hold up the peer review process. 6. Please include your full ethics statement in the ‘Methods’ section of your manuscript file. In your statement, please include the full name of the IRB or ethics committee who approved or waived your study, as well as whether or not you obtained informed written or verbal consent. If consent was waived for your study, please include this information in your statement as well. 7. Please include captions for your Supporting Information files at the end of your manuscript, and update any in-text citations to match accordingly. Please see our Supporting Information guidelines for more information: http://journals.plos.org/plosone/s/supporting-information. 8. If the reviewer comments include a recommendation to cite specific previously published works, please review and evaluate these publications to determine whether they are relevant and should be cited. There is no requirement to cite these works unless the editor has indicated otherwise.

**Additional Editor Comments:**

Please follow the reviewers' comments to revise the manuscript.

Reviewers' comments:

Reviewer's Responses to Questions

**Comments to the Author**

1. Is the manuscript technically sound, and do the data support the conclusions?

Reviewer #1: Partly

Reviewer #2: Yes

2. Has the statistical analysis been performed appropriately and rigorously? 

Reviewer #1: No

Reviewer #2: Yes

3. Have the authors made all data underlying the findings in their manuscript fully available?

Reviewer #1: Yes

Reviewer #2: Yes

4. Is the manuscript presented in an intelligible fashion and written in standard English?

Reviewer #1: Yes

Reviewer #2: Yes

5. Review Comments to the Author

**Reviewer #1:** This manuscript addresses an important and persistent challenge in health services research: the prediction of 30-day readmissions following pneumonia hospitalization. The authors present a rigorously conducted comparison of traditional logistic regression with more advanced machine learning approaches (XGBoost and deep neural networks) using rich longitudinal EMR data that includes continuous clinical variables. The work is methodologically sound, clearly written, and transparent in reporting negative findings—an important contribution given publication bias toward performance gains.

While the primary hypothesis that advanced ML models would outperform logistic regression was not supported, the manuscript offers meaningful insights into why performance plateaus occur and identifies novel predictors (notably BUN and substance abuse). The study is valuable, but several conceptual, methodological, and interpretive issues should be addressed to strengthen its impact and clarity.

A. Major Comments

1. Conceptual Framing: “Nonlinearity” vs. “Predictive Ceiling”

Strength:

The rationale for exploring nonlinear relationships using ML models is well articulated and grounded in the literature. The post hoc spline analysis of BUN is a strong and thoughtful addition.

Concern:

The manuscript continues to frame the study primarily as an evaluation of whether nonlinear modeling improves prediction, even though the results clearly suggest a predictive ceiling driven by limitations of EMR data rather than model choice.

Recommendation:

Reframe the manuscript more explicitly around the concept of data-limited prediction, rather than model-limited prediction. This would better align the Introduction, Results, and Discussion. Consider stating earlier (in the Introduction or Methods) that an alternative goal is to characterize what ML models learn differently, even when performance is similar.

2. Feature Pre-Selection and Its Impact on Machine Learning Models

Strength:

The marginal screening procedure is clearly described and appropriately adjusted for multiple testing.

Concern:

The use of univariate screening (p<0.10 with BH correction) prior to ML modeling may constrain the ability of ML models to discover interactions and nonlinear effects, undermining the core premise of the study.

This is particularly relevant for DNNs and XGBoost, which are designed to handle high-dimensional feature spaces and correlated predictors.

Recommendation:

Explicitly acknowledge this as a limitation in the Discussion.

Clarify why marginal screening was preferred over embedded feature selection methods (e.g., L1 regularization, tree-based importance).

Consider adding a sensitivity analysis (if feasible) comparing models with and without pre-screening, or at least discuss how this choice may have attenuated ML advantages.

3. Interpretation of SHAP Values for Binary Predictors

Strength:

The use of SHAP values enhances model interpretability and is a major strength of the paper.

Concern:

The approach of calculating SHAP values only among patients with endorsed binary predictors (to avoid washout) is unconventional and may inflate perceived importance for rare variables (e.g., drug abuse at 1.1%).

Recommendation:

Provide a stronger methodological justification for this approach.

Discuss how this decision affects comparability across predictors and models.

Consider reporting prevalence-adjusted SHAP summaries or including a sensitivity comparison using standard SHAP aggregation.

4. Readmission Outcome Definition and Noise

Strength:

The authors appropriately align the outcome definition with CMS quality measures.

Concern:

All-cause readmission is acknowledged as noisy, but the implications are understated. Given that pneumonia-related factors account for a minority of readmissions, the models may be penalized for predicting events driven by post-discharge social or behavioral factors absent from the EMR.

Recommendation:

Expand the Discussion on outcome heterogeneity and label noise.

Explicitly consider whether pneumonia-specific readmissions (even as a secondary analysis) might yield different insights.

Frame the modest AUROC/AUPRC as partly a consequence of outcome misalignment with available predictors.

5. Clinical Translation and Actionability

Strength:

The manuscript thoughtfully discusses implementation challenges and resource allocation.

Concern:

Despite modest PPV, the manuscript stops short of proposing a clear operational use case for the models (e.g., triage vs. screening vs. layered intervention).

Recommendation:

Strengthen the Discussion by explicitly stating:

What clinical decision this model should support.

Whether it is best suited for rule-out, risk stratification, or resource prioritization.

How clinicians should interpret a “high-risk” label given the PPV of ~0.16.

B. Minor Comments

1. Abstract

Consider clarifying that performance was similar rather than simply “did not improve,” to avoid implying methodological failure.

The phrase “novel predictors” may overstate novelty; consider “previously underemphasized predictors.”

2. Methods

Clarify whether continuous variables were normalized or transformed prior to DNN training.

Provide additional detail on DNN architecture (layers, activation functions) in the main text or supplement.

3. Results

Table 2 would benefit from explicitly stating that confidence intervals overlap substantially across models.

Consider adding calibration metrics (e.g., calibration slope or Brier score), as these are clinically relevant even when AUROC is modest.

4. Figures

Figure 2 is strong conceptually; adding a brief clinical annotation (e.g., normal BUN range) directly on the plot would enhance interpretability.

Ensure consistent terminology: “SHaply” → “SHapley” in Figure 1 caption.

5. Language and Style

The manuscript is well written; minor tightening is possible in the Discussion to reduce repetition around EMR limitations.

Consider reducing redundancy between Discussion paragraphs on social determinants of health.

**Reviewer #2:** It is a excellent study in the field of pneumonia. It will helps the physician treating pneumonia and can help them identifying patients of pneumonia which are at mores risk for readmission. This knowledge can help them reducing risk for readmission and help in keeping patients out of hospital.

6. PLOS authors have the option to publish the peer review history of their article (what does this mean?). If published, this will include your full peer review and any attached files.

Reviewer #1: No

Reviewer #2: No

---

## [Author Response · Author response to Decision Letter 1]

22 Apr 2026

Response to Reviewers

Reviewer 1

A. Major Comments

1. Conceptual Framing: “Nonlinearity” vs. “Predictive Ceiling.”

Strength:

The rationale for exploring nonlinear relationships using ML models is well articulated and grounded in the literature. The post hoc spline analysis of BUN is a strong and thoughtful addition.

Concern:

The manuscript continues to frame the study primarily as an evaluation of whether nonlinear modeling improves prediction, even though the results clearly suggest a predictive ceiling driven by limitations of EMR data rather than model choice.

Recommendation:

Reframe the manuscript more explicitly around the concept of data-limited prediction, rather than model-limited prediction. This would better align the Introduction, Results, and Discussion. Consider stating earlier (in the Introduction or Methods) that an alternative goal is to characterize what ML models learn differently, even when performance is similar.

Author response (AR): Thank you for your suggestion. We agree that highlighting data limitations in the introduction would create a more consistent narrative. However, we still want to keep our main hypotheses regarding model limitations. Therefore, we have added a discussion of data limitations in our introduction on lines 84-87.

“One reason for this similarity may be that these models excluded continuous variables, which could have benefited from nonlinear or interaction effects (e.g., vital signs or laboratory findings). Given that the data available in medical records (required for real-time prediction in healthcare settings) is generally limited, discovering these additional relationships could be even more important for model development.”

2. Feature Pre-Selection and Its Impact on Machine Learning Models

Strength:

The marginal screening procedure is clearly described and appropriately adjusted for multiple testing.

Concern:

The use of univariate screening (p<0.10 with BH correction) prior to ML modeling may constrain the ability of ML models to discover interactions and nonlinear effects, undermining the core premise of the study.

This is particularly relevant for DNNs and XGBoost, which are designed to handle high-dimensional feature spaces and correlated predictors.

Recommendation:

Explicitly acknowledge this as a limitation in the Discussion.

Clarify why marginal screening was preferred over embedded feature selection methods (e.g., L1 regularization, tree-based importance).

Consider adding a sensitivity analysis (if feasible) comparing models with and without pre-screening, or at least discuss how this choice may have attenuated ML advantages.

AR: We acknowledge this point. We considered the trade-off between screening (to reduce overfitting to the training data) and using the full dataset (to enable machine learning models to detect interactions). In response, we added a post-hoc analysis that reruns our models without screening on lines 273-283.

“To help improve the generalizability of our models, we carried out an initial marginal screening process, which involves a tradeoff because it may miss variables that influence importance only through interactions. Since XGBoost and DNN models can evaluate feature importance natively and still consider variables important only in interactions, we reran our primary analyses without the marginal screening process (Supplement 4).

In this post hoc analysis, we observed that the performance of all models generally stayed the same, although their metrics were slightly worse. The logistic regression model was affected the most, with the average AUROC decreasing from 0.63 to 0.59, and the percentage of patients with readmissions in the bottom decile of risk scores increasing from 4% to 9%.”

We also now discuss this limitation on lines 367-372.

“Finally, we lightly screened our features to help trained models better generalize to the testing data. When removing this step in our post hoc analysis, it seemed particularly important for the logistic regression model, while the advanced ML models showed little change. We recognize that more sophisticated screening methods exist that could have affected our results, which remains a limitation of this work.”

3. Interpretation of SHAP Values for Binary Predictors

Strength:

The use of SHAP values enhances model interpretability and is a major strength of the paper.

Concern:

The approach of calculating SHAP values only among patients with endorsed binary predictors (to avoid washout) is unconventional and may inflate perceived importance for rare variables (e.g., drug abuse at 1.1%).

Recommendation:

Provide a stronger methodological justification for this approach.

Discuss how this decision affects comparability across predictors and models.

Consider reporting prevalence-adjusted SHAP summaries or including a sensitivity comparison using standard SHAP aggregation.

AR: Thank you; we understand your concern. We provided current SHAP summaries to highlight the importance of less common yet critical diagnosis codes that are significant for clinical judgment. SHAPs in this context mainly reflect model coefficients from logistic regression (conditional importance). However, we recognize that global importance, which includes prevalence, would also be valuable. In response, we expand our explanation for conditional SHAPs on lines 173-177.

“For binary predictors, we calculated SHAP values within patients who had that variable endorsed to highlight clinically important diagnoses. Most of these diagnoses occur only in a small subset of patients (e.g., <10%). Their significance would be diminished or “washed out” by low prevalence in overall SHAP value calculations due to zero contributions from most patients.”

We also provide results summarizing global SHAP values on lines 227-234 and in Supplementary File 3.

“Given that we calculated binary variable SHAP values only within patients who endorsed that variable, we then recalculated global SHAP values without this restriction (Supplement 3). As expected, almost all diagnoses dropped out of the list of most important predictors and were replaced by continuous variables that were present in all patients. This was an exception for employment status in the logistic regression model and for fluid/electrolyte disorders in the XGBoost model. The top predictors within the DNN model remained largely unchanged since they were already primarily continuous variables.”

Lastly, we revised lines 296-299.

“All of our models identified substance abuse as an important predictor of readmission among the few patients with this diagnosis code (1.1%); for the overall sample, vitals and laboratory findings were most important based on global SHAP values.”

4. Readmission Outcome Definition and Noise

Strength:

The authors appropriately align the outcome definition with CMS quality measures.

Concern:

All-cause readmission is acknowledged as noisy, but the implications are understated. Given that pneumonia-related factors account for a minority of readmissions, the models may be penalized for predicting events driven by post-discharge social or behavioral factors absent from the EMR.

Recommendation:

Expand the Discussion on outcome heterogeneity and label noise.

Explicitly consider whether pneumonia-specific readmissions (even as a secondary analysis) might yield different insights.

Frame the modest AUROC/AUPRC as partly a consequence of outcome misalignment with available predictors.

AR: This is an excellent point. In response, we provide additional discussion on lines 357-364.

“Third, our outcome measure—all-cause readmission—includes admissions unrelated to the index pneumonia hospitalization, which may introduce noise into our models. For example, some patients may be readmitted for unrelated injuries (e.g., falls, accidents) rather than for pneumonia sequelae. In such cases, admission characteristics reflecting pneumonia severity would not be informative. Consequently, the modest performance metrics (e.g., AUROC, AUPRC) may be partly attributable to a misalignment between admission characteristics and unrelated readmissions in some patients.”

5. Clinical Translation and Actionability

Strength:

The manuscript thoughtfully discusses implementation challenges and resource allocation.

Concern:

Despite modest PPV, the manuscript stops short of proposing a clear operational use case for the models (e.g., triage vs. screening vs. layered intervention).

Recommendation:

Strengthen the Discussion by explicitly stating:

What clinical decision this model should support.

Whether it is best suited for rule-out, risk stratification, or resource prioritization.

How clinicians should interpret a “high-risk” label given the PPV of ~0.16.

AR: We agree that this deserves some elaboration, although it is difficult to provide a definitive guideline when such models are not currently deployed (i.e., unknown resource allocation, workflow integration, etc). In response, we have expanded our discussion to include more details on lines 344-351.

“Although we are hesitant to provide specific guidelines for implementation since these algorithms are not currently used by any hospitals and logistics such as resource allocation and workflow integration remain uncertain, these models could still be used for both risk stratification and resource allocation. For example, labeling nearly half of the sample as high risk (39-43% at the Youden Index) resulted in PPVs between 0.15 and 0.17. These patients could have brief discharge conversations. Labeling 10% of the sample resulted in PPVs between 0.19 and 0.20. These patients could also receive follow-up phone calls.”

B. Minor Comments

1. Abstract

Consider clarifying that performance was similar rather than simply “did not improve,” to avoid implying methodological failure.

The phrase “novel predictors” may overstate novelty; consider “previously underemphasized predictors.”

AR: Thanks, done. We also modified line 294, which mentioned novel.

2. Methods

Clarify whether continuous variables were normalized or transformed prior to DNN training.

Provide additional detail on DNN architecture (layers, activation functions) in the main text or supplement.

AR: We have added that we normalized variables in Supplement 2. We already provide extensive details in Supplement 2 on the architecture, layer purposes, and activation functions. We would be happy to provide more details if you could specify them, please.

3. Results

Table 2 would benefit from explicitly stating that confidence intervals overlap substantially across models.

AR: This makes sense. We have added these details on lines 209-211.

“Our more advanced ML models performed similarly, providing comparable average metrics and mostly overlapping confidence intervals across all models.”

Consider adding calibration metrics (e.g., calibration slope or Brier score), as these are clinically relevant even when AUROC is modest.

AR: Good point. In response, we have added this information to the Methods section (lines 165-167), Results section (211-216), and Table 2.

“To further evaluate discrimination and calibration, we visually examined calibration plots and calculated Brier scores (0=perfect calibration and discrimination).”

“When visually inspecting calibration plots along with Brier scores, logistic regression seemed to have the best calibration according to the Brier score, although all models performed relatively similarly. Notably, from calibration plots (data not shown), all models consistently predicted higher probabilities of readmission than actually observed across the risk spectrum.”

4. Figures

Figure 2 is strong conceptually; adding a brief clinical annotation (e.g., normal BUN range) directly on the plot would enhance interpretability.

AR: Good idea. Added.

Ensure consistent terminology: “SHaply” → “SHapley” in Figure 1 caption.

AR: Thank you for this suggestion. Fixed.

5. Language and Style

The manuscript is well written; minor tightening is possible in the Discussion to reduce repetition around EMR limitations.

Consider reducing redundancy between Discussion paragraphs on social determinants of health.

AR: We agree. In response, we have significantly reduced this section on lines 319-325. We also removed the paragraph related to treatment planning for patients with drug abuse, as it only affected a few patients.

“However, the consistent performance of our models and other published studies suggests that the modest predictive accuracy may be due to the inherent limitations of the data available in electronic medical records. EMRs lack key social determinants of health (e.g., housing stability, health literacy, lifestyle habits, environmental factors), which can affect readmission risk. Therefore, collecting some of this information during the intake process could improve the algorithms.”

Reviewer #2: It is a excellent study in the field of pneumonia. It will helps the physician treating pneumonia and can help them identifying patients of pneumonia which are at mores risk for readmission. This knowledge can help them reducing risk for readmission and help in keeping patients out of hospital.

---

## [Decision Letter · Decision Letter 1]

5 May 2026

Leveraging Nonlinear Relationships and Interactions to Improve 30-Day Pneumonia Readmission Machine Learning Models

PONE-D-26-02589R1

Dear Dr. Mortensen,

We’re pleased to inform you that your manuscript has been judged scientifically suitable for publication and will be formally accepted for publication once it meets all outstanding technical requirements.

Kind regards,

Oliver Schildgen

Academic Editor

PLOS One

Additional Editor Comments (optional):

Reviewers' comments:

Reviewer's Responses to Questions

**Comments to the Author**

1. If the authors have adequately addressed your comments raised in a previous round of review and you feel that this manuscript is now acceptable for publication, you may indicate that here to bypass the “Comments to the Author” section, enter your conflict of interest statement in the “Confidential to Editor” section, and submit your "Accept" recommendation.

Reviewer #1: All comments have been addressed

2. Is the manuscript technically sound, and do the data support the conclusions?

Reviewer #1: (No Response)

3. Has the statistical analysis been performed appropriately and rigorously? 

Reviewer #1: (No Response)

4. Have the authors made all data underlying the findings in their manuscript fully available?

Reviewer #1: (No Response)

5. Is the manuscript presented in an intelligible fashion and written in standard English?

Reviewer #1: (No Response)

6. Review Comments to the Author

Reviewer #1: (No Response)

7. PLOS authors have the option to publish the peer review history of their article (what does this mean?). If published, this will include your full peer review and any attached files.

Reviewer #1: No

---

## [Editor Report · Acceptance letter]

PONE-D-26-02589R1

PLOS One

Dear Dr. Mortensen,

I'm pleased to inform you that your manuscript has been deemed suitable for publication in PLOS One. Congratulations! Your manuscript is now being handed over to our production team.

Kind regards,

on behalf of

Professor Oliver Schildgen

Academic Editor

PLOS One